# The Potential of Acorn Extract Treatment on PUFAs Oxidative Stability: A Case Study on Fish Cooking Wastewater

**DOI:** 10.3390/foods13060935

**Published:** 2024-03-19

**Authors:** Helena Araújo-Rodrigues, Tânia Bragança Ribeiro, Manuela Machado, Carlos D. Pereira, Manuela E. Pintado

**Affiliations:** 1CBQF—Centro de Biotecnologia e Química Fina—Laboratório Associado, Escola Superior de Biotecnologia, Universidade Católica Portuguesa, Rua Diogo Botelho 1327, 4169-005 Porto, Portugal; hrodrigues@ucp.pt (H.A.-R.); tribeiro@ucp.pt (T.B.R.); mmachado@ucp.pt (M.M.); 2Instituto Politécnico de Coimbra, Escola Superior Agrária, 3045-601 Coimbra, Portugal; cpereira@esac.pt

**Keywords:** acorn shell, antioxidant extract, PUFA oxidation, fish byproducts, sardine cooking effluent

## Abstract

Fish byproducts are valuable sources of Ω-3 polyunsaturated fatty acids (PUFAs). Their valorization potentially alleviates pressure on this sector. This study uses a circular economy approach to investigate the oil fraction from sardine cooking wastewater (SCW). Analysis of its fatty acid (FA) profile revealed promising PUFA levels. However, PUFAs are highly susceptible to oxidation, prompting the exploration of effective and natural strategies to replace synthetic antioxidants and mitigate their associated risks and concerns. An antioxidant extract from acorn shells was developed and evaluated for its efficacy in preventing oxidative degradation. The extract exhibited significant levels of total phenolic compounds (TPC: 49.94 and 22.99 mg TAE or GAE/g DW) and antioxidant activities (ABTS: 72.46; ORAC: 59.60; DPPH: 248.24 mg TE/g DW), with tannins comprising a significant portion of phenolics (20.61 mg TAE/g DW). LC-ESI-UHR-QqTOF-MS identified ellagic acid, epicatechin, procyanidin B2 and azelaic acid as the predominant phenolic compounds. The extract demonstrated the ability to significantly reduce the peroxide index and inhibit PUFA oxidation, including linoleic acid (LA), eicosapentaenoic (EPA), and docosahexaenoic acid (DHA). This approach holds promise for developing stable, functional ingredients rich in PUFAs. Future research will focus on refining oil extraction procedures and conducting stability tests towards the development of specific applications.

## 1. Introduction

Fish is widely recognized as a valuable nutrient source for the human diet due to its bioactive molecules associated with several health properties, including antioxidant, anti-inflammatory, neuroprotective, and hepatoprotective effects [1,2]. Remarkably, fish and other marine animals are excellent sources of long-chain polyunsaturated fatty acids (PUFAs) [1,2,3,4]. Fatty acids (FAs) can be categorized based on the presence and number of double bonds, namely, saturated (SFA; absence of double bonds), monounsaturated (MUFA; single double bond), and PUFAs (multiple double bonds). PUFAs are further classified based on the position of the first bond on the methyl terminal. These are categorized as omega-3 (Ω-3) and omega-6 (Ω-6) if the first double bond is found in the third or sixth position from the methyl terminal, respectively [5,6,7].

Fatty fishes such as sardines, salmon, mackerel, trout, and herring possess a rich composition of PUFAs [3], mainly eicosapentaenoic (EPA), docosahexaenoic (DHA), and docosapentaenoic (DPA) acids. These are pivotal for brain, cardiovascular, and kidney functions, as well as inflammatory and immunological responses [1,7,8]. Through epidemiological and controlled clinical studies, the European Food Safety Authority (EFSA) recommends a daily intake of at least 250 mg of EPA and DHA to support normal heart function in healthy adults [5].

Despite the essential role of PUFAs in maintaining human body homeostasis, they cannot be synthesized endogenously [6,7]. Consequently, an increase in the market of dietary supplements and functional ingredients containing EPA and DHA has been reported [1,6]. The rising demand for fisheries’ production of food, nutraceuticals, pharmaceuticals, cosmetics, and other applications, coupled with pollution and environmental changes, has exerted significant pressure on marine ecosystems [6,9]. Therefore, finding alternative sources of PUFAs is a highly relevant topic.

Estimates suggest that for some types of fish and processing methods, 70% of the processed fish results in byproducts [10,11]. These byproducts, encompassing bone, frame, skin, internal organs, and muscle, represent a rich reservoir of PUFAs. These byproducts hold high potential in numerous sectors, particularly the food industry [2,4,10]. The waste generated during the fish canning process has also been recognized as a rich source of countless bioactive compounds such as lipids, proteins, bio-polymers, and minerals. Fish cooking wastewater represents a source of significant amounts of proteins and Ω-3 [4,12]. Valorizing these byproducts aligns with the principles of a circular economy approach and helps mitigate the technical, economic, and environmental issues associated with their elimination [9,10,12,13].

However, PUFAs’ chemical nature, characterized by multiple double bonds, makes them highly unstable and easily oxidizable [6,8,14,15], generating off-flavours [6,8,14]. Furthermore, oxidation products (e.g., hydroperoxides and aldehydes) have also been associated with harmful health effects. Accordingly, there is growing interest in value-added strategies to stabilize fish oil PUFAs, such as the application of antioxidant extracts [6,14]. Preference for natural antioxidants over synthetic ones is driven by their potential health benefits [6,15] and possible undesired effects, as well as restricted usage of synthetic antioxidants in some countries [3]. 

Among natural antioxidant sources, extracts from edible plants such as rosemary, sesame seed, and green tea [15,16] are examples of antioxidant extracts with high potential in the prevention of lipid oxidation due to their rich composition of bioactive compounds (e.g., carotenoids, polyphenols, and tocopherols) [6,13,16]. Notably, acorn, the fruit of the *Quercus* spp., is also rich in phytochemicals, such as phenolic compounds and sterols. *Quercus* spp. is widely abundant in temperate areas, corresponding to approximately 34% of the forest area in Portugal. Although acorn has been extensively applied in animal feeding, over 50% of its production is wasted [17,18]. However, its biological potential and high nutritional value highlight acorns’ interest in the human diet [19,20]. Acorns are a good source of carbohydrates (83–86%) and protein (more than 7%), with a low content of fat (less than 5%) [17,18]. Antioxidant, cardioprotective, and anti-carcinogenic are acorns’ most reported biological properties [18,19,20].

Recent studies have identified promising antioxidant activities and rich phytochemical compositions in acorn fruit, cup, and shell [18,21]. The shell comprises approximately 20% of the acorn’s weight and is typically discarded or underutilized [18]. These findings have led to innovative approaches suggesting that acorn shell extract avoids oxidation of aromas extracted from sardine cooking wastewater (SCW), mainly aldehyde oxidation. These studies developed an SCW valorization approach, creating value for aromas present in this effluent for feed and aquaculture applications [9,22]. In other work, SCW was also studied to valorize its protein and bioactive fractions for food, pharmaceutical, or cosmetic fields. The acorn shell extract was also applied due to its antioxidant nature [23]. However, neither study has evaluated the bioactive profile of acorn shell extract and its antioxidant properties. In addition, this is the first study where the antioxidant potential of shell acorn extract is explored to prevent PUFA oxidation (during sardine oil separation from SCW).

This innovative study investigated the oxidative stability potential of a natural antioxidant extract on a fish processing byproduct (a rich source of PUFAs). Specifically, a shell acorn extract was developed and applied to the fat fraction of SCW for the first time. The acorn shell extract was extensively characterized, including assessment of antioxidant capacity, total phenolic compounds (TPC) and tannin concentration, using different methods. The main bioactive molecules in this antioxidant extract were also identified and quantified. The total fat, fatty acid (FA) profile and oxidation index were tested and compared with non-treated SCW. This study aimed to decipher the impact of acorn shell extract on PUFA oxidation.

## 2. Materials and Methods

### 2.1. Chemicals and Raw Materials 

SCW was kindly provided by A Poveira S.A. (Porto, Portugal). Herdade do Freixo do Meio (Montemor-o-Novo, Portugal) supplied acorn shell (*Quercus* spp.), where cork oak forest is essentially made up of oaks—holm oaks (*Quercus rotundifolia*), cork oaks (*Q. suber*), carrascos (*Q. coccifera*), cerquinho oaks (*Q. faginea*) and black oaks (*Q. pyrenaica*). 

For antioxidant and TPC methods, ABTS diammonium salt (2,2′-azino-bis(3-ethylbenzothiazoline-6-sulphonic acid)), 2,2-diphenyl-1-picrylhydrazyl, fluorescein, 2,2′-azo-bis-(2-methylpropionamidine)-dihydrochloride (≥97%; AAPH), sodium carbonate, gallic acid (≥99%), 6-hydroxy-2,5,7,8-tetramethylbroman-2-carboxylic acid (≥97%; Trolox), tannic acid and polyvinylpyrrolidone (PVPP) were supplied by Sigma-Aldrich (St. Louis, MO, USA), while Folin–Ciocalteu was supplied by Merck (Darmstadt, Germany). Organic HPLC grade solvents for LC-ESI-UHR-QqTOF-MS analysis were also obtained from Sigma-Aldrich. 

Regarding FA derivatization and analysis, HPLC grade solvents were used, namely, methanol, hexane, acetonitrile and dimethylformamide (DMF), and were purchased from VWR Scientific (VWR chemicals, Karlsruhe, Germany). Sulphuric acid and sodium methoxide were obtained from Honeywell (Charlotte, NC, USA) and Acros Organics (Geel, Belgium), respectively. The internal standard used for fatty acid quantification was Tritridecanoin (33-1300-13), supplied by Larodan Research Grade Lipids (Solna, Sweden), while Supelco 37 Component FAME Mix was obtained from Sigma-Aldrich. For total fat and peroxide index analysis, hydrochloric acid (32%), sodium thiosulfate, potassium iodide and starch were purchased from Merck, while glacial acetic acid (99.8%), petroleum ether was from VWR Scientific.

### 2.2. Antioxidant Extract Preparation

Based on a comprehensive literature review, the authors chose a water extraction method to obtain the acorn shell extract for this investigation. This approach has been widely utilized in numerous previous studies [9,18,22,23,24,25]. 

The acorn shell was processed by milling using a grinder (Coffee Grinder TAURUS Aromatic II, Barcelona, Spain). The moisture content of the resulting acorn shell powder was determined by drying it in an oven (Memmert, Schwabach, Germany) at 105 °C until reaching a constant weight. To prepare the antioxidant extract, the milled sample was mixed with deionized water at a concentration of 8% (*w*/*v*). Subsequently, the mixture was incubated at 100 °C for 15 min and filtered throughout filter paper Whatman No 1. The resulting acorn shell extracts were prepared in triplicate and stored at −80 °C until further application. 

### 2.3. Acorn Shell Extract Characterization

#### 2.3.1. Total Phenolic Compounds (TPC)

To quantify the TPC present in acorn shell extract, the Folin–Ciocalteau colourimetric method was used as described by Araújo-Rodrigues et al. [13]. The reaction was carried out in a 96-well microplate using the mixture of 30 µL of extract with 100 µL of Folin–Ciocalteu reagent (20% *v*/*v*), followed by the addition of 100 µL of sodium carbonate (7.4% *w*/*v*). Simultaneously, a standard curve was prepared using various concentrations of gallic acid (ranging from 0.015 to 0.225 mg/mL) and subjected to the same reaction conditions. The reaction proceeded in darkness for 1 h at room temperature. Then, a multidetection plate reader (Synergy H1, Winooski, VT, USA), operated using the Gen5 Biotek software version 3.04, was used to monitor the absorbance at 750 nm. Each extract was analyzed in triplicate. TPC values were expressed as milligrams of gallic acid equivalent (GAE) per 100 g of dry weight (DW) and milligrams of tannic acid equivalent (TAE) per 100 g of dry weight (DW).

#### 2.3.2. Total Tannins

The Tannin concentration was determined using insoluble polyvinyl-polypirrolidone (PVPP), following the method described by Rakić [26]. Briefly, 1 mL of extract was combined with 100 mg PVPP and vortexed. Then, the mixture was incubated for 15 min at 4 °C. Subsequently, the mixture was centrifuged for 10 min at 3000 rpm. The clear supernatant was collected for quantification, following the procedure described in the previous subsection. The supernatant corresponds to non-tannin phenolics, as tannins bind to PVPP. Finally, the tannin concentration was calculated using the difference between total phenolics and non-tannin phenolics (mg TAE/g of acorn shell DW).

#### 2.3.3. Antioxidant Capacity

The antioxidant capacity was evaluated using ABTS, DPPH, and ORAC. The ABTS assay established by Coscueta et al. [27] was used to evaluate the antioxidant potential of acorn shell extract. The reaction was performed in a 96-well microplate with 180 µL of ABTS•+ working solution and 20 µL of extract. A Trolox standard curve, ranging from 25 and 175 µM, was also prepared in the same proportion. The ABTS•+ working solution was prepared on the day of the experiment by 0.45 µm filtration of ABTS•+ stock solution and adjusting the absorbance to 0.70 ± 0.02 with up water. The ABTS•+ stock solution was prepared earlier by reacting potassium persulfate (2.45 mM) with ABTS•+ (7 mM) in ultra-pure (UP) water for 16 h in darkness at room temperature. After mixture and incubation in darkness for 5 min at room temperature, the absorbance was measured at 734 nm with a multidetection plate reader. Each extract was analyzed in triplicate. The results were expressed as mg of Trolox Equivalent (TE) per 100 g of DW.

For the DPPH assay, the method described by Schaich et al. [28] was followed to evaluate the hydrophilic antioxidant compounds in acorn shell extract. The reaction occurred in a 96-well microplate with 25 μL of extract and 175 μL of DPPH working solution. A Trolox standard curve (concentrations ranging from 25 to 175 μM) was prepared in the same proportion. The working solution was prepared daily from a DPPH stock solution (600 μM), diluted to approximately 60 μM, and the absorbance was adjusted at 515 nm to 0.600 ± 0.100. After mixing and incubating in darkness for 30 min at 25 °C, the absorbance was measured in a multidetection plate reader at 515 nm. The antioxidant results were also expressed as mg of TE per 100 g of DW.

The ORAC assay was conducted in a multidetection plate reader, according to the methodology outlined by Coscueta et al. [27]. The reaction occurred in black polystyrene 96-well microplates (Nunc, Roskilde, Denmark), combining 120 µL of fluorescein (final concentration of 70 nM) with 20 µL of antioxidant extract. A blank and Trolox standard curve were included as well. The antioxidant extract was prepared in phosphate buffer (75 mM; pH 7.4). The mixture was then incubated for 10 min at 37 °C. Subsequently, 60 µL of APPH (final concentration of 12 mM) was added, and the microplate was mixed. The mixture was incubated for 80 min at 37 °C. The fluorescence readings were taken every minute, with excitation and emission wavelengths of 485 and 538 nm, respectively. The results were expressed in µmol TE by 100 g of DW.

### 2.4. Analysis of Phenolic Compounds by Liquid Chromatography-Electrospray Ionization Quadrupole Time-of-Flight Mass Spectrometry (LC-ESI-UHR-QqTOF-MS)

The identification of the phenolic compounds present in the extract was carried out with an LC-ESI-UHR-QqTOF-MS system (Bruker Daltonics, Billerica, MA, USA) following the methodology of Monforte et al. [29], with some modifications. The adjustments were in the gradient elution program (mobile phase A: 0.1% aqueous formic acid and mobile phase B: acetonitrile with 0.1% formic acid) to obtain a good separation of phenolic compounds, namely: 0–5 min (5% B), 5–25 min (15% B), 25–35 min (30% B), 35–40 min (95% B), 40–41 (5% B) and 41–42 min (0% B). 

Post-acquisition internal mass calibration used sodium formate clusters, with the sodium formate delivered by a syringe pump at the start of each chromatographic analysis. High-resolution mass spectrometry was used to identify the compounds. The elemental composition for the compound was confirmed according to accurate mass and isotope rate calculations designated mSigma (Bruker Daltonics). The accurate mass measurement was within 5 mDa of the assigned elemental composition, and mSigma values of <20 provided confirmation. Compounds were identified based on their accurate mass [M − H]. Three independent analyses were performed in each of the triplicate extracts obtained. Before the injection, a solid phase extraction (SPE) of the acorn shell extract samples (3 mL) was done to remove interferences that cause high background, misleading peaks, and poor sensitivity during chromatographic analysis. Sep-pak C18 were conditioned with 10 mL of methanol, followed by 5 mL of MilliQ water. After adjusting to pH 6.0 with 0.1 mol/L HCl, sample loading was done, followed by washing with 1 mL of 5% (*v*/*v*) MeOH aqueous solution. 

### 2.5. Application of Antioxidant Extract on Sardine Cooking Wastewater (SCW) 

This SCW byproduct is obtained from sardine steaming for 7 min at 100 °C in cooking chambers. The resulting condensed water (SCW) is collected and immediately refrigerated. One fraction was treated with 1% acorn extract (*v*/*v*), while the other served as a control (untreated). To recover the fat fraction, the wastewater was heated to 35 °C and centrifuged in a centrifugal separator Westfalia type ADB at 6500× *g* (Westfalia Separator AG, Oeld/Westfalia, Oelde, Germany). The fat fraction was immediately frozen at −25 °C until further analysis. The defatted fraction of SCW was submitted to microfiltration and ultrafiltration to allow the recovery of other solids, namely proteins [23,30] and aromas [9,22], which were then subjected to different treatments.

#### 2.5.1. Total Fat and Oxidation Index

The total fat content was determined for control (non-treated) and antioxidant extract-treated samples by the Soxhlet method based on NP1613:1979 [31]. The results were expressed as g per 100 g of sample. Initially, a hydrolysis step with 4 N hydrochloric acid was conducted, followed by petroleum ether extraction. The oxidation index was also evaluated using the peroxide index method based on International Standard Operation (ISO) 3960:2007 [32], with results expressed in milliequivalents of oxygen (meq O_2_) by kg of fat.

#### 2.5.2. Fatty Acid (FA) Profile 

The fatty acid profile was determined by derivatization, followed by gas chromatography, coupled with a flame ionization detector (GC-FID), according to the methodology of Pimentel et al. [33]. In the derivatization process, 100 mL of tritridecanoin (1.50 mg/mL) was added as an internal standard to 150 mg of SCW (control) or SCW treated with acorn extract. A sequential addition of 2.26 mL of methanol, 1 mL of hexane and 240 µL of sodium methoxide (prepared in 5.4 M of methanol) was carried out. After vortex mixing, the mixture was incubated at 80 °C for 10 min, followed by cooling in ice and the addition of DMF (1.25 mL) and sulphuric acid (1.25 mL; 3 M). The samples were mixed and incubated again at 60 °C for 30 min. After cooling, 1 mL of hexane was added, and the samples were vortexed and centrifuged (1250× *g*, 18 °C, and 5 min). The fatty acid methyl esters (FAMEs) were collected in the upper layer for GC-FID analysis.

FAMEs were analyzed in a gas chromatograph HP6890A (Hewlett-Packard, Avondale, PA, USA) equipped with a flame ionization detector (GLC-FID) and a BPX70 capillary column (60 m × 0.25 mm × 0.25 μm; SGE Europe Ltd., Courtaboeuf, France). The equipment running conditions were as follows: the injector 250 °C, split 25:1, injection volume of 1 µL, detector 275 °C, and hydrogen was used as carrier gas (20.5 psi). Oven temperature conditions were as follows: start at 60 °C (held for 5 min), then raise at 15 °C/min to 165 °C (held for 1 min), and finally at 2 °C/min to 225 °C (held for 2 min). Finally, a Supelco 37 Component FAME Mix was used to identify FA. The fatty acids present in this mix are: C4:0 (Butryic); C6:0 (Caproic), C8:0 (Caprylic), C10:0 (Capric), C11:0 (Undecanoic), C12:0 (Lauric), C13:0 (Tridecanoic), C14:0 (Myristic), C14:1 (Myristoleic), C15:0 (Pentadecanoic), C15:1 (cis-10-Pentadecenoic), C16:0 (Palmitic), C16:1 (Palmitoleic), C17:0 (Heptadecanoic), C17:1 (cis-10-Heptadecenoic), C18:0 (Stearic), C18:1n9c (Oleic), C18:1n9t (Elaidic), C18:2n6c (Linoleic), C18:2n6t (Linolelaidic), C18:3n6 (γ-Linolenic), C18:3n3 (α-Linolenic), C20:0 (Arachidic), C20:1n9 (cis-11-Eicosenoic), C20:2 (cis-11,14-Eicosadienoic), C20:3n6 (cis-8,11,14-Eicosatrienoic), C20:3n3 (cis-11,14,17-Eicosatrienoic), C20:4n6 (Arachidonic), C20:5n3 (cis-5,8,11,14,17-Eicosapentaenoic), C21:0 (Henicosanoic), C22:0 (Behenic), C22:1n9 (Erucic), C22:2 (cis-13,16-Docosadienoic), C22:6n3 (cis-4,7,10,13,16,19-Docosahexaenoic), C23:0 (Tricosanoic), C24:0 (Lignoceric), and C24:1n9 (Nervonic). Response factors, limits of detection (LOD) and limits of quantification (LOQ) were calculated using GLC-Nestlé36. The LOD was 0.79 ng FA/mL, and the LOQ was 2.64 ng FA/mL.

### 2.6. Statistical Analysis

Statistical analysis was performed using SPSS statistical software (28.0). After confirmation of normal data distribution, the means were compared by Student’s *t*-test, with a degree of significance of *p* < 0.05. 

## 3. Results and Discussion

In this study, the acorn shell extract was chemically characterized, and its oxidative stabilization potential was evaluated on the fat fraction of SCW for the first time.

### 3.1. Acorn Shell Extract Characterization

#### 3.1.1. Total Phenolic Compounds, Total Tannins and Total Antioxidant Activity 

The TPC, tannin and non-tannin concentrations in acorn shell extract and its antioxidant potential were investigated in the first phase. These values are presented in Table 1. 

Concerning the TPC of the acorn shell, the concentration was 50.29 mg GAE/g of its DW and 22.99 mg TAE/g of its DW. A high presence of phenolic compounds may explain the antioxidant potential and capacity of acorn shell extract to improve the shelf life and resistance to oxidation of PUFAs. The double bonds present in PUFAs are highly unstable and easily oxidizable [5,7,12,13], so developing effective antioxidant strategies is crucial [5,13]. Natural antioxidants have gained popularity compared to synthetic ones due to their health-promoting properties [5,11,13], lack of side effects, and lack of usage restrictions [3]. Several natural antioxidants have been tested for stabilizing edible oils rich in PUFAs. Some examples include plant extracts rich in carotenoids (e.g., β-carotene), tocopherols (e.g., α- and γ-tocopherol), vitamins (e.g., ascorbic acid), flavonoids (e.g., quercetin), catechins (e.g., epigallocatechin) and phenolic acids (e.g., gallic acid) [14].

An interesting phytochemical composition and bioactive properties have been associated with acorn and its byproducts, such as shells and cups. Several studies have shown strong antioxidant activity mainly due to its high total phenolic content. As previously mentioned, the acorn shell extract showed potential in the stabilization of aromas [9,22] and of the bioactive fraction (mainly composed of proteins and lipids) of SCW [23]. 

Youn et al. [16] also evaluated acorn shell extracts’ TPC and antioxidant potential. The authors demonstrated that extracts of acorn shells may be a promising source of antioxidants with anti-obesity activity. Water extract exhibited a TPC concentration of 375.96 mg GAE/g of extract, lower than the acorn shell extract prepared in the present study (419.03 ± 23.89 mg GAE/g of DW of extract). Although the extraction temperature was similar in both cases, Youn et al. [16] extraction time was double (30 min), and the acorn shells belong to *Quercus acutissima* species produced in Korea. Increasing the extraction time may improve the extraction yield of phenolic compounds. However, exposure to higher temperatures (100 °C) for extended periods may also be damaging. 

In a more recent study, several solvents were tested to develop acorn shell extract as a component for active food packaging [34]. The water extract showed an even lower TPC value (80.65 mg GAE/g of extract). The species of acorn shells used were not indicated, but the geographical localization (holm oak wood pastures in Valle de los Pedroches, Cordoba, Spain) was mentioned. The variability in plant species and geographic location may be the main reason for the differences observed in the TPC of acorn shell extracts across studies.

A water extraction method was chosen to obtain the acorn shell extract in this study based on the methodologies reported in the literature [9,18,22,23,24,25]. Some authors demonstrated that water-soluble phenolics may impact antioxidant potential more than methanol-soluble ones [18]. Besides that, boiling water extraction is the primary and traditional method for acorn oil production [21], and to overcome the issue of tannins. Tannins and some flavonoids are the most prevalent free phenolics of acorns. They are considered anti-nutritional macromolecules, astringent to taste and toxic when consumed in high amounts [17]. Consequently, beyond efficient extraction, boiling water extraction may have the additional advantage of decreasing tannin concentration. However, the results of tannins (20.61 mg TAE/g of its DW) suggested that most of the free phenolic compounds present in *Quercus* spp. acorns are tannins, and a small fraction corresponds to non-tannins (2.39 mg TAE/g of its DW), which aligns with literature reports for acorns [17,26,35,36]. Additionally, aqueous extracts without organic solvents are safer, less expensive and can be easily applied to food matrices at the industrial level, implying low costs [37]. 

Makhlouf et al. [33] assessed the phenolic and antioxidant properties of acorn flour and oil extracted from *Quercus ilex* L. and *Quercus suber* L. species. TPC ranged from 19.50 to 146.4 mg GAE/100 g of DM, flours of both species exhibited higher TPC concentration than oil extracts. Other authors prepared several *Quercus suber* extracts using different solvents (water, methanol and hexane) and plant parts (acorns and leaves) and compared the TPC, antioxidant potential and phenolic profile. The highest TPC was found in methanolic extracts and the highest concentrations in plant leaves (around 211.0 mg GAE/g of extract), while in acorn extract, the concentrations were approximately 49.0 mg GAE/g of extract. Water extracts also exhibited considerable TPC concentrations, namely, 61.2 and 17.1 mg GAE/g of extract in leaves and acorns, respectively [24]. Accordingly, the present TPC results aligned with these studies, although the phytochemical composition generally differs among several plant parts.

ABTS scavenging activity is extensively used to screen the antioxidant properties of natural sources such as plants, fruits and byproducts. ABTS scavenging activity of acorn shells was approximately 72.46 mg TE/g of DW of acorn shells. The results of the ORAC assay suggested an antioxidant capacity of 248.24 mg TE/g of DW of acorn shell. Both assays measure the influence of amphipathic and hydrophilic compounds on antioxidant capacity [13]. However, ABTS and ORAC assays possess distinct underlying mechanisms. Concerning DPPH, it evaluates the contribution of lipophilic compounds to antioxidant capacity [13], and the results indicated an antioxidant capacity of 59.60 mg TE/g of DW. 

As previously pointed out, in other studies targeting acorn shell extracts, water extracts exhibited higher antioxidant capacities than methanol extracts, suggesting that higher antioxidant capacities result from a higher concentration of phenolic compounds [18]. However, Custódio et al. [22] concluded that methanolic extracts of acorns possessed significantly higher antioxidant potential than water extracts, which is expected since methanol is considered one of the best solvents to maximize the phenolic compounds extracts. The distinct plant variety, phytochemical composition of acorns and shells and different extraction times may also contribute to contradictory conclusions. Makhlouf et al. [38] also determined the antioxidant capacity using ABTS and DPPH methods. The antioxidant values varied from 1.27 and 44.5 mg TE/g DM for the ABTS assay, while in the DPPH assay, flour extract exhibited a higher antioxidant capacity of approximately 52.62 mg TE/g DM. In both assays, the results aligned with the present antioxidant results.

Regarding antioxidant activity measured by ORAC, acorn shell extract exhibited a significant value (2177.79 ± 320.51 mg TE/g of DW of extract). ORAC antioxidant activity determination was only found in the literature on acorn flour extracts after dehulling and using methanol as solvent [17] and in water and ethanol: water acorn cup extracts [39]. Aqueous and 50% ethanol acorn cup extracts exhibited the highest ORAC antioxidant values, with approximately 1100 and 1800 mg TE/g extract, respectively [39].

#### 3.1.2. Identification of Phenolic Compounds by Liquid Chromatography-Electrospray Ionization Quadrupole Time-of-Flight Mass Spectrometry (LC-ESI-UHR-QqTOF-MS)

The principal phenolic compounds identified in acorn shell extract and their peaks are shown in Table 2. Ellagic acid is the predominant compound identified in acorn shell extract (Figure 1a). Similar results were described for acorns [17,40] and acorn cups [39]. The highest amount of ellagic acid associated with the highest antioxidant activity by ORAC of 50% ethanol acorn cup extract was developed by Yin et al. [34]. More recently, Başyiğit et al. [41] tested the food-protecting ability of an ethanolic extract of *Quercus infectoria* gall, where ellagic acid was the predominant compound. This extract was applied in pasteurized milk, reducing total bacterial and yeast mold counts. In addition, high antimicrobial activity against *Escherichia coli* was noted. However, lipid oxidation was not evaluated in this study.

Epicatechin is the second most abundant phenolic compound detected in acorn shell extract (Figure 1b). Epicatechin has been reported to effectively reduce lipid oxidation and potentially reduce the formation of advanced lipid oxidation end products in a fish oil oxidation model [42]. This compound was among the most abundant compounds quantified in the free phenolic extract of acorn (*Quercus variabilis* Blume) kernel, as well as azelaic acid, which was also detected in significant quantity in our study [43]. Ellagic acid was also noticed in significant amounts but in the bound phenolic extract. 

**Table 2 foods-13-00935-t002:** Phenolic compounds tentatively identified in acorn shell extract by LC-ESI-UHR-QqTOF-MS.

	RT (min)	*m*/*z* Experimental	Molecular Formula	*m*/*z* Calculated	Error (ppm)	mSigma Value	MS^2^ Fragments	Proposed Compound	Peak Area	Reference
1	9.5	577.1342	C_30_H_26_O_12_	577.1351	1.6	13.0	407.0763 (87.40%), 339.0871 (16.10%), 289.0170 (100%), 245.0820 (24.54%), 125.0243 (55.57%)	B-type Procyanidin	52,832	[44]
2	9.8	137.0245	C_7_H_6_O_3_	137.0244	−0.3	16.8	136.0166 (40.70%), 119.0441 (24.91%), 108.0215 (100%), 91.0184 (18.95%), 81.0371 (22.81%)	Protocathecuic aldeheide	10,252	[45]
3	11.5	289.0715	C_15_H_14_O_6_	289.0718	1.0	2.9	221.0816 (36.83%), 218.9045 (59.31%), 151.0393 (70.13%), 137.0242 (54.39%), 123.0452 (84.58%), 109.0297 (100%)	Epicatechin isomer	30,937	[43]
4	11.8	577.1347	C_30_H_26_O_12_	577.1351	0.8	10.6	409.0772 (70.09%), 289.0715 (100%), 137.0231 (11.36%), 125.0242 (48.04%)	B-type Procyanidin	83,285	[44]
5	13.0	289.0715	C_15_H_14_O_6_	289.0718	0.7	11.2	151.0399 (39.08%), 139.0410 (14.33%), 125.0243 (22.77%), 123.0451 (100.00%) 109.0289 (78.25%), 97.0291 (19.51%)	Epicatechin	102,286	[43]
6	14.8	635.0888	C_27_H_24_O_18_	635.0890	0.2	13.3	465.0679 (95.16%), 313.0567 (79.60%), 169.0138 (100.00%), 125.0248 (15.55%)	Trigalloylglucose isomer	19,664	[40]
7	27.2	300.9992	C_14_H_6_O_8_	300.9990	−0.7	3.8	300.9984 (100.00%), 283.9961 (18.57%), 229.0137 (18.34%), 185.0244 (12.30%)	Ellagic acid	166,609	[40]
8	30.0	187.0973	C_9_H_16_O_4_	187.0976	1.4	1.1	135.9145 (22.35%), 125.0964 (59.03%), 123.0802 (100.00%)	Azelaic acid	75,217	[43]
9	34.8	201.1129	C_10_H_18_O_4_	201.1132	1.6	3.4	184.1083 (29.22%), 139.1116 (100%), 11.0803 (28.81%)	Unknown compound	32,788	
10	36.1	593.1295	C_30_H_26_O_13_	593.1301	1.0	6.8	285.0400 (100.00%)	Tiliroside	47,255	[46]

Tiliroside and B-type procyanidins were also detected in acorn shell extract. Gallic acid has been reported as the main phenolic compound in acorn shells [18]. However, gallic acid was not detected in our extract. The results indicated that the acorn shell extract contained many low-molecular-weight phenolics such as ellagic acid, epicatechin, and azelaic acid. 

### 3.2. Assessment of the Stabilization of Polyunsaturated Fatty Acids (PUFAs) Using Acorn Shell Extract: A Case Study of Sardine Cooking Wastewater (SCW)

To evaluate the potential of the acorn shell extract in PUFA oxidative stability, the SCW treated with this extract was compared with the untreated SCW (control sample) regarding total fat content, oxidation index and fatty acid profile. The total fat content and oxidation indices are described in Table 3. As expected, both samples showed a similar total fat content, approximately 83% (Table 3). SCW is an effluent generated during fish processing, which possesses a rich composition of proteins and peptides, aldehydes, and Ω3 PUFA [4,9,22,30]. Sardine is a fat-rich fish [3] and as expected, its cooking wastewater is rich in fat. Pereira et al. [9] characterized some of the physical and chemical parameters of this SCW. These parameters were approximately a pH of 6.5, an organic load of 28 mg/L, and total protein and lipid contents of 25.38 and 28.13 mg/L, respectively. The processing approach applied to SCW concentrated the fat fraction, with lipids as the most prevalent group. 

Regarding the oxidation index through peroxide index, both SCW samples presented high peroxide values (PVs), i.e., superior to the PV regulatory limit for edible oils of 10 mEq/kg and exceeding the limit of 5 mEq/kg according to the standards for fish oils [47]. However, several studies showed that oil from byproducts could be successfully refined to improve its overall quality, enabling its further application as edible oil, feed additive, or even for human consumption as a fish oil supplement [48,49]. So, a refinement process should be applied to achieve a good quality fish oil from SCW. 

Despite the high PV of both SCW samples, the results indicated a significant decrease in the oxidation index from 98.75 to 97.50 meq O_2_/kg of fat in the SCW with acorn shell extract application. Acorn extracts from fruit, leaves and wood chips were also applied to inhibit lipid oxidation (TBARS/PV) in different foods, namely meat, soybean and sunflower oils [50]. For example, a *Q. branti* acorn extract with a lower TPC value (226.4 mg GAE/g of extract DW) than the acorn shell extract developed in this work exhibited a similar positive effect on the PV decrease. A reduction of the PV value of 12.5 meq O_2_/kg of fat to 10 meq O_2_/kg of fat was detected and maintained during the first 10 days of storage at 60 °C [51]. 

More recently, an acorn shell extract was tested as a component for active food packaging [34]. This study incorporated an acorn shell extract from ultrasound-assisted extraction (70% acetone:30% water, 30 min) into chitosan and polyvinyl alcohol (PVA) films into different concentrations. Rincón et al. [31] verified that films with higher concentrations of acorn shell extract (5 and10%) exhibited the capacity to reduce lipid oxidation (measured by TBARS methodology) of packaged soybean oil. Moreover, a higher antioxidant activity measured by DPPH was noticed with increased acorn shell extract concentration. In conclusion, the phenolic compounds present in acorn and acorn shell extract seem to have the capacity to retard the oxidation reactions due to their ability to scavenge free radicals, chelate metal ions and act as reducing agents [52]. 

To understand the impact of antioxidant extract application on oxidative stability, FA were identified and quantified (Table 4). Analyzing FA composition allowed us to estimate the extent of oil oxidation and assess the protective effect of acorn shell extract against FA oxidation. Since the composition of SCW undergoes oxidation changes, PUFAs are transformed into MUFAs and SFAs, and *cis* FAs into *trans*-FAs [52,53].

Both samples possess a high diversity of FA, and several significant differences were found in the different FA quantified. The most prevalent FAs present in the samples were myristic acid (C14), palmitic acid (C16), palmitoleic acid (C16:1 c9), oleic acid (C18:1 c9), C17:1 c9 and EPA (C20:5 c5c8c11c14c17), with a concentration between approximately 45.52 and 137.63 µg/mg in control sample and varying between 47.20 and 137.27 µg/mg. 

The FA profile suggested significant variations in saturated and unsaturated fatty acid concentrations between control and acorn extract-treated samples. These variations were more evident in PUFAs, namely linoleic acid (LA; C18:2 c9c12), C18:2 c9c15, EPA (C20:5 c5c8c11c14c17) and DHA (C22:6 c4c7c10c13c16c19). The PUFAs were significantly more prevalent in acorn extract-treated samples. For example, the results suggested that in the antioxidant-treated sample, the EPA had a concentration of 115.7 µg/mg and DHA of 28.4 µg/mg, in contrast to 102.7 and 0.0 µg/mg in the control sample. Fish oil is generally highly unstable due to the significant presence of unsaturated fat [16], as corroborated by this study.

Although no considerable differences were found between SFAs and MUFAs, significant differences in PUFAs were found between the control sample (SCW) and SCW treated with acorn extract. Thus, the results demonstrate that acorn shell extract can potentially protect FAs against oxidation, namely retarding the oil oxidation, since the conversion of double bonds present in PUFAs to single bonds (hydrogenation) is less pronounced in the antioxidant-treated sample [52]. This protective effect of acorn shell extract may be due to its phenolic compounds, such as ellagic acid, epicatechin, and azelaic acid, identified by the chromatographic analyses above. 

Analysis of the FA profile of these essential PUFAs suggests the possibility that acorn shell extract prevents the oxidation of EPA and DHA, which is very promising in terms of nutritional profile. Consequently, these first results indicated that acorn extract might play a key role in the oxidative stability of PUFAs in the fat fraction of SCW.

## 4. Conclusions

The high demand for fisheries in the food, nutraceutical, pharmaceutical, and cosmetics industries exerts significant pressure on the fish industry. The consumption of fish-derived supplements has increased due to their rich Ω-3 PUFAs nutritional composition and associated health claims. In the fish industry, byproducts constitute a substantial fraction of processed fish (about 70%). The fish canning industry generated numerous byproducts with significant concentrations of valuable molecules. One example is fish cooking wastewater that possesses a high concentration of proteins and Ω-3. Their elimination has significant economic and environmental implications. Thus, valorizing SCW may be an effective way to develop high-value products within a circular economy approach, thereby contributing to sustainability and alleviating pressure in the fish sector. Consequently, in the present work, SCW was investigated for its potential food applications as a rich source of PUFAs. 

One of the primary challenges associated with PUFA utilization in food and other fields is their high susceptibility to oxidation. This challenge may be addressed through the application of synthetic and natural antioxidants. Natural antioxidants, in addition to exhibiting high antioxidant activity, offer other health benefits, are highly acceptable by the consumer, and are typically not associated with adverse effects or usage restrictions. Accordingly, an acorn shell antioxidant extract was developed and, for the first time, tested on fish cooking wastewater oxidative regression. The extensive characterization demonstrated an interesting TPC, antioxidant potential and significant presence of ellagic acid, epicatechin, B-type procyanidin and azelaic acid. 

The FA profile suggested a very interesting PUFA profile in SCW. The results also indicated that acorn shell extract might reduce the peroxide index and prevent PUFA oxidation. The SCW with acorn shell extract treatment possesses high concentrations of Ω-3 and Ω-6. Some examples of PUFAs are LA, EPA, DPA and DHA. Thus, this innovative approach may correspond to an environmentally friendly alternative for developing a functional ingredient rich in PUFA and resistant to oxidation. Future research should focus on refining the SCW process to enhance its peroxide value and conducting stability tests to understand PUFA stability under specific processing food conditions (e.g., temperature and storage conditions).

## Figures and Tables

**Figure 1 foods-13-00935-f001:**
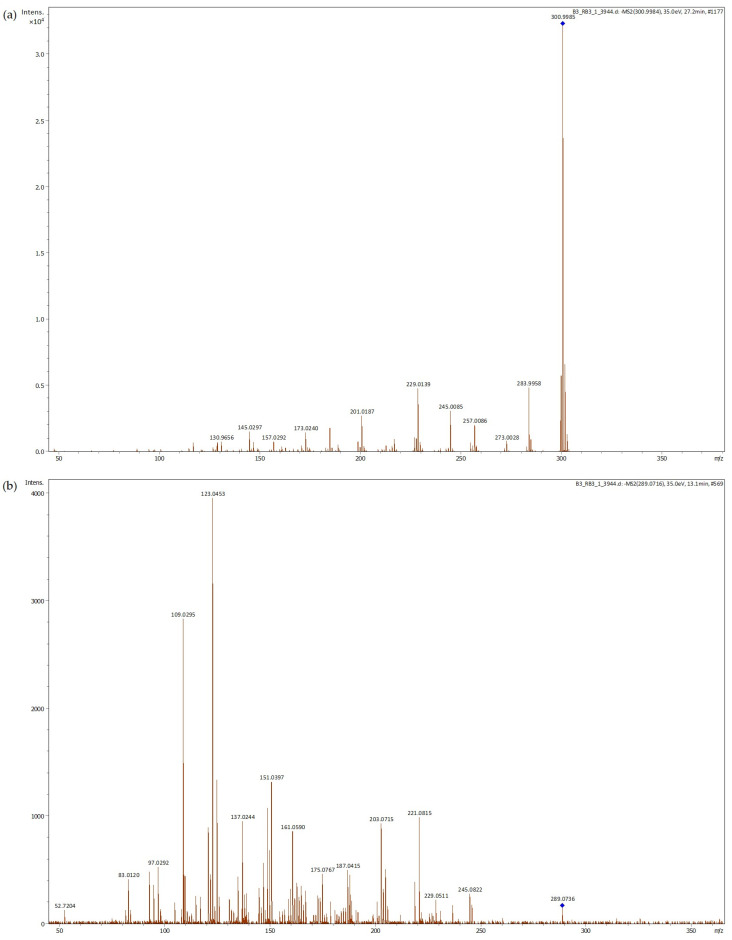
LC-ESI-UHR-QqTOF-MS of acorn extract *m*/*z* 300.9992 ((**a**), tentatively identified ellagic acid) and *m*/*z* 289.0715 ((**b**), tentatively identified epicatechin).

**Table 1 foods-13-00935-t001:** Total phenolic compounds, tannin and non-tannin concentration and antioxidant potential of acorn shell extract.

	*Total Phenolic Concentration (Follin)*	
Units	mg gallic acid equivalent (GAE)/ g of dry weight (DW) of acorn shell	mg gallic acid equivalent (GAE)/ g of dry weight (DW) of extract
Phenolics	50.29 ± 1.57	419.03 ± 23.89
Units	mg tannic acid equivalent (GAE)/ g of dry weight (DW) of acorn shell	mg tannic acid equivalent (GAE)/ g of dry weight (DW) of extract
Phenolics	22.99 ± 1.31	205.37 ± 12.56
	*Tannin concentration*	
Units	mg tannic acid equivalent (TAE)/ g of dry weight (DW) of acorn shell	mg tannic acid equivalent (TAE)/ g of dry weight (DW) of extract
Tannins	20.61 ± 1.21	184.04 ± 11.67
Non-tannins	2.39 ± 0.17	21.33 ± 1.48
	*Antioxidant potential*	
Units	mg Trolox equivalent (TE)/g of dry weight (DW) of acorn shell	mg Trolox equivalent (TE)/ g of dry weight (DW) of extract
ABTS	72.46 ± 3.25	608.61 ± 34.65
DPPH	59.60 ± 4.27	493.21 ± 27.74
ORAC	248.24 ± 27.75	2177.79 ± 320.51

ABTS—2,2′-azino-bis(3-ethylbenzothiazoline-6-sulfonic acid); DPPH—1,1-difenil-2-picrilhidrazil; ORAC—Oxygen Radical Absorbance Capacity.

**Table 3 foods-13-00935-t003:** Total fat content and peroxide index.

Sample	Fat Content	Peroxide Index
SCW—control ^1^	83.10 ± 0.14 ^a^	98.75 ± 0.07 ^a^
SCW—acorn shell extract treated ^2^	83.25 ± 0.28 ^a^	97.50 ± 0.14 ^b^

SCW—sardine cooking wastewater. ^1^ g/100 g fresh weight; ^2^ meq O_2_/kg of fat. Different superscript letters in the same column are significantly different (*p* < 0.05).

**Table 4 foods-13-00935-t004:** Fatty acid profile (µg/mg of fresh weight of SCW) present in sardine cooking wastewater (control sample) and sardine cooking wastewater treated with acorn shell extract.

Fatty Acid	SCW (Control)	SCW with Acorn Shell Extract Treatment
Lauric acid—C12:0	0.63 ± 0.05 ^a^	0.65 ± 0.00 ^a^
Isomyristic acid—C14:0	0.12 ± 0.01 ^a^	0.12 ± 0.01 ^a^
Myristic acid—C14:0	45.52 ± 4.71 ^a^	47.20 ± 0.36 ^a^
Pentadecenoic acid—C15:0	3.03 ± 0.34 ^a^	3.02 ± 0.02 ^a^
Isopentadecanoic acid—C15:0 i	0.04 ± 0.00 ^a^	0.04 ± 0.00 ^a^
C15:1	0.64 ± 0.26 ^a^	0.53 ± 0.06 ^a^
Palmitic acid—C16	137.63 ± 16.55 ^a^	137.27 ± 0.72 ^a^
Trans fatty acid—C16:1 t9	2.08 ± 0.23 ^a^	2.04 ± 0.01 ^a^
C16:1 c7	0.14 ± 0.00 ^a^	0.13 ± 0.02 ^a^
Palmitoleic acid—C16:1 c9	70.10 ± 7.03 ^a^	73.01 ± 2.29 ^a^
Isoheptadecanoic acid—C17 i	0.99 ± 0.10 ^a^	1.02 ± 0.00 ^a^
C16:1 c11	2.88 ± 0.00 ^a^	4.64 ± 0.48 ^b^
Margaric acid—C17:0	8.31 ± 0.50 ^a^	8.98 ± 0.06 ^b^
C17:1 c9	N.D.	0.96 ± 0.02
C17:1 c10	8.31 ± 0.50 ^a^	9.46 ± 0.13 ^b^
Isooctanoic acid—C18 i	0.24 ± 0.02 ^a^	0.26 ± 0.00 ^b^
Stearic acid—C18:0	28.57 ± 3.76 ^a^	26.88 ± 0.12 ^a^
C17:1 t10	0.12 ± 0.01 ^a^	0.12 ± 0.01 ^a^
C18:1 t12	0.84 ± 0.01 ^a^	0.84 ± 0.01 ^a^
Oleic acid—C18:1 c9	101.05 ± 31.34 ^a^	87.65 ± 0.38 ^a^
C18:1 c11	N.D.	23.59 ± 3.97 ^a^
C18:1 c12	0.72 ± 0.09 ^a^	0.65 ± 0.01 ^a^
C18:1 c13	0.24 ± 0.09 ^a^	0.16 ± 0.02 ^a^
Linolelaidic acid—C18:2 t9t12	3.34 ± 0.34 ^a^	3.42 ± 0.01 ^a^
C18:2 c9t12	N.D.	0.15 ± 0.00
Linoleic acid—C18:2 c9c12 (LA)	5.37 ± 0.05 ^a^	5.97 ± 0.62 ^b^
C18:2 c9c15	0.07± 0.01 ^a^	3.98 ± 0.03 ^b^
C18:3 c6c9c13	1.70 ± 0.15 ^a^	1.51 ± 0.01 ^a^
α-Linolenic acid—C18:3 c9c12c15 (ALA)	4.51 ± 0.56 ^a^	4.45 ± 0.09 ^a^
Arachidic acid—C20:0	N.D.	13.61 ± 0.12
C18:2 c0t11	27.95 ± 2.59 ^b^	14.04 ± 0.01 ^a^
c20:1 c9	4.45 ± 0.47 ^a^	4.25 ± 0.18 ^a^
Arachidonic acid—C20:4 c5c8c11c14	5.77 ± 0.48 ^a^	5.96 ± 0.10 ^a^
Behenic acid—C22:0	10.29 ± 1.09 ^a^	9.61 ± 0.03 ^a^
Eicosapentaenoic acid—C20:5 c5c8c11c14c17 (EPA)	102.66 ± 7.01 ^a^	115.72 ± 0.74 ^b^
Lignoceric acid—C24:0	0.98 ± 0.04 ^a^	0.89 ± 0.00 ^b^
Docosapentaenoic acid—C22:5 (DPA)	10.21 ± 0.95 ^a^	10.93 ± 0.01 ^a^
Docosahexaenoic acid—C22:6 c4c7c10c13c16c19 (DHA)	N.D.	28.39 ± 0.15
Total quantified FAs	640.95 ± 68.83 ^a^	688.10 ± 1.74 ^b^
Σ SFA	236.16 ± 19.62 ^a^	249.06 ± 0.90 ^a^
Σ MUFA	190.61 ± 28.36 ^a^	206.97 ± 1.70 ^a^
Σ PUFA	161.51 ± 8.50 ^a^	194.52 ± 0.34 ^b^

SCW—sardine cooking wastewater. FAs—fatty acids. Σ SFA—sum of saturated fatty acids. Σ MUFA—sum of unsaturated fatty acids. Σ PUFA—sum of polyunsaturated fatty acids. N.D. —Not detected. Different superscript letters in the same line are significantly different (*p* < 0.05).

## Data Availability

The original contributions presented in the study are included in the article, further inquiries can be directed to the corresponding author.

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
