# Peer review of "The Potential of Acorn Extract Treatment on PUFAs Oxidative Stability: A Case Study on Fish Cooking Wastewater"

_foods, 2024, doi:10.3390/foods13060935_

Round 1

Reviewer 1 Report

Comments and Suggestions for Authors

Major comments

In this article, the author uses acorn extracts to improve the stability of oils extracted from by-products of fish processing. The author has identified the main components in the acorn extract. The author found that acorn extract can significantly alter the fatty acid composition of oils. This is a meaningful attempt, and some interesting phenomena have been reported. However, the research is relatively rudimentary, and some research data have not been well explained. At the same time, the data does not sufficiently support the main conclusions. The author needs to carefully check and confirm their experimental data.

Specific comments

2.2. Antioxidant extract preparation needs to rewrite.

The table 1 needs to be remade. Also, there are no units in the table.  

The author believes that the tannins in the extracts are harmful. It is necessary for the author to test the tannin content of the extracts from the species.

3.1. Acorn shell extract characterization. It is best to divide it into two parts.

The author used the extract in the extraction of oils and fats. However, the author did not discuss the impact of the addition of extracts on the efficiency of fat extraction.

After collection, the SCW was immediately refrigerated, and one fraction was treated with 1% acorn extract (v/v). How is this addition amount determined?

The author believes that the extract can stabilize polyunsaturated fatty acids. But the content of monounsaturated fatty acids such as oleic acid (C18:1) has significantly increased. Interestingly, the content of another type of fatty acid (C17:1) has significantly decreased. How to explain this? The author needs to carefully check and confirm their experimental data.

 There are errors in the following part:

(SFA; absence of double bounds), unsaturated (UFA; single double bound),line 36

There are some grammatical errors in the article, and the author needs to check carefully, the following are just examples:

Even though the high applicability of PUFAs, the chemical nature of their multiple bonds makes them highly unstable and easily oxidizable [5,7,12,13] generating off-fla

vours in foods [5,7,12].

docosahexaenoic acids (DHA), line 42

PUFAs are also classified by the first bond position on the me, line 37

ones is registered due to their potential health benefits, line 68

strong antioxidant properties, line 82

Shell is approximately 20% of acorn weight, typically discardedor underutilized, line 83

theSCW, line 205

difference in the studies' TPC of water acorn shell extracts. Line 293

3.2. Evaluation of acorn shell extract PUFA oxidative stabilization – SCW case study. Line 369

Comments on the Quality of English Language

Extensive editing of English language required

Author Response

Please find attached a Word file containing the final version of the manuscript. We would like to thank the reviewers for their suggestions and comments, which were very important and valuable and greatly contributed to improving the quality and clarity of the manuscript. The new modifications were highlighted in red to facilitate the revision. We hope you find it adequate for publication in Foods in its present form.

Yours sincerely,

Manuela E. Pintado, on behalf of all authors

Reviewer 2 Report

Comments and Suggestions for Authors

Foods (foods-2894300). “The Potential of Acorn Extract Treatment on PUFAs Oxidative Stability: a Case Study on Fish Cooking Wastewater”.

Reviewer comments:

In general, the present manuscript: “The Potential of Acorn Extract Treatment on PUFAs Oxidative Stability: a Case Study on Fish Cooking Wastewater” by Araújo-Rodrigues and collaborators focused to investigate the oxidative stability potential of a natural antioxidant extract on a fish processing by product (a rich source of PUFAs), and it has an interesting goal. Some specific comments are given bellow, to help improving the quality of the manuscript reviews.

Final remark: this manuscript needs a minor revision. Below are some comments to improve the quality of the manuscript.

General comments

- Usually, key words are words that do not contain in the title of the manuscript. Review the entire of manuscript.

- Currently, most of the scientific manuscript are presented as hypotheses to be more attractive and interesting than description of goals. The present manuscript can be presented with hypothesis. We suggest the authors to present this manuscript with more attractive hypothesis and to make the manuscript more interesting.

- Conclusions must be made based on the results obtained and in relation to the objectives and hypotheses (both in the abstract and at the end of the manuscript – conclusion section).

Are there any reviews on sardine cooking wastewater? A more plausible justification for the use of this wastewater needs to be presented in the introduction.

A doubt. Does the manuscript have no discussion? Or would it be Results and Discussion? If this is the latter case, you missed writing "discussion" in the subtitle (it's just results). I don't know if it's the journal's policy, but generally the results are described separately from the discussion.

Table numbers are wrong. Correct the entire manuscript.

Author Response

Please find attached a Word file containing the final version of the manuscript. We would like to thank the reviewers for their suggestions and comments, which were very important and valuable and greatly contributed to improving the quality and clarity of the manuscript. The new modifications were highlighted in blue to facilitate the revision. We hope you find it adequate for publication in Foods in its present form.

Yours sincerely,

Manuela E. Pintado, on behalf of all authors

Reviewer 3 Report

Comments and Suggestions for Authors

The work presented has merit for publication in Foods journal. It deals both with the exploitation of sardine cooking wastewater that still contains bioactive compounds, which should be protected from oxidation; hence, here comes the role of acorn shells extract as protectants of this oxidation. I have limited comments that if considered, will further improve the manuscript prior. 

1) With regard to the peroxide index and the impact of the extract in this direction in need to point out, that despite the statistical significance, still the decrease is not suffcient. Do the authors have another explanation for this observation (apart from the differences in the quantified FAs and prevention of oxidation of some of them)? Additional standardization in the provision of the extract could benefit (lower) the peroxide index. 

2) The section 3, should be renamed to Resutls and Discussion. Then, please renumber the section Conclusions. 

3) At the beginning of section 3, I suggest moving the first two paragraphs after or inside section 3.1. These are more discusion than results, and is essential to begin this section with the obtained results.

4) In Table 2, either you use only "tentatively identified" or if you used authentic standards for identification please provide the info (and in the materials and methods for chemicals). Please consider providing one to two figures with MS/MS fragments of compounds tentatively identified. It would add value to the manuscript. 

5) Minor editing comments: Page 1, line 45, correct to "epidemiological". Page 7, line 321, hydrophilic "what"? Compounds? Please complete the sentence.  

Author Response

(The authors gave the same response as above.)
